# Mode attraction, rejection and control in nonlinear multimode optics

Kunhao Ji [1] ✉, Ian Davidson [1], Jayanta Sahu[1], David J. Richardson[1,2], Stefan Wabnitz [3] & Massimiliano Guasoni [1] ✉

Novel fundamental notions helping in the interpretation of the complex dynamics of nonlinear systems are essential to our understanding and ability to exploit them. In this work we predict and demonstrate experimentally a fundamental property of Kerr-nonlinear media, which we name mode rejection and takes place when two intense counter-propagating beams interact in a multimode waveguide. In stark contrast to mode attraction phenomena, mode rejection leads to the selective suppression of a spatial mode in the forward beam, which is controlled via the counter-propagating backward beam. Starting from this observation we generalise the ideas of attraction and rejection in nonlinear multimode systems of arbitrary dimension, which paves the way towards a more general idea of all-optical mode control. These ideas represent universal tools to explore novel dynamics and applications in a variety of optical and non-optical nonlinear systems. Coherent beam combination in polarisation-maintaining multicore fibres is demonstrated as example.

Multimode waveguides have become one of the major research topics of the last decade in optics. The initial interest in multimode optical fibres for next-generation optical communication systems[1,2] has rapidly spread to a diversity of applications, including astronomy[3], high-speed datacentres[4], imaging[5] and more recently integrated photonics[6], which attests to the relevance of this field within the broad photonics community[7].

The massive interest in multimode fibres has triggered the investigation of nonlinear effects in these systems. Whether it represents an issue to avoid or to exploit for high-power lasers[8] and optical communications[9], nonlinear dynamics in multimode waveguides undoubtedly represents a rapidly growing research field. Indeed, nonlinear coupling among different spatial modes—mediated by cross-phase modulation, four-wave mixing and Raman scattering- and its interplay with temporal dispersion gives rise to complex, multi-faceted dynamics where much is still unknown[10–15].

Over the last few years novel scenarios have been disclosed, including wideband supercontinuum mediated by intermodal nonlinear interactions[16,17], nonlinear conversion via intermodal four-wave-mixing[18,19], spatio-temporal multimode solitons[20] and mode-locking[21], pulse combining[22] and spatial beam self-cleaning[13,23], to name a few. All the above-mentioned phenomena share a co-propagating geometry, whereas counter-propagating geometries in multimode waveguides have so far been left largely unexplored.

Nonlinear counter-propagating systems have been extensively studied in single-mode waveguides since the '80 s, where new fundamental results have been demonstrated, along with relevant applications. This includes spatial/temporal chaos and bistability[24–27] as well as the existence of special polarisation states that play the role of robust attractor states.

The notion of attraction plays a central role in mathematics and physics. The existence of system attractors has been observed in different kind of optical fibres, from isotropic[28], to highly birefringent[29,30] up to randomly birefringent fibres[31]. While different types of fibre exhibit different kinds of attractors, the underlying attraction process is characterised by a similar dynamic[32]: independently of its input

[1]Optoelectronics Research Centre, University of Southampton, Southampton SO17 1BJ, United Kingdom. [2]Microsoft (Lumenisity Limited), Unit 7, The Quadrangle, Abbey Park Industrial Estate, Romsey SO51 9DL, United Kingdom. [3]Dipartimento di Ingegneria dell'Informazione, Elettronica e Tele-comunicazioni, Sapienza University of Rome, 00184 Rome, Italy. ✉e-mail: k.ji@soton.ac.uk; m.guasoni@soton.ac.uk

polarisation state, a forward signal (FS) is attracted towards an attractor state that is fixed by a counter-propagating backward control beam (BCB), launched at the opposite fibre end. The attraction process is driven by the Kerr-nonlinear interaction between the FS and BCB. As a result, for an increasing degree of nonlinearity, the attraction becomes more effective and robust to external perturbations (Fig. 1a, b).

These two features—robustness and independence of the input conditions—are peculiarities of the counter-propagating setup, whereas they are not present in standard co-propagating systems[33,34]. Besides the fundamental interest, these outcomes have been exploited to implement a variety of novel optical devices, including lossless all-optical polarisers[35], all-optical polarisation scramblers[36] and nonlinear focusing mirrors for temporal compression[37].

Differently from single-mode waveguides, counter-propagating setups in multimode waveguides have been little addressed so far. The existence of mode attractors in a fibre supporting two spatial modes has been envisaged[34,38,39]. However, this process has been measured and quantified for the first time only recently in our work ref. 39. Moreover, there has not been any attempt to analyse the most general case of multimode waveguides supporting any number of modes. This is arguably due to the additional complexity intrinsic to the counter-propagating multimode dynamics, both in terms of fundamental understanding as well as of the experimental setup.

Nevertheless, the richness of recent results in nonlinear multimode waveguides, along with earlier outcomes in single-mode fibres with counter-propagating configurations as depicted above, let us to anticipate a variety of nonlinear phenomena in counter-propagating multimode systems. The scope of this paper is to initiate research in this direction.

Here, we generalise the idea of mode attraction in waveguides supporting an arbitrary number N of modes. Moreover, we predict and

demonstrate experimentally a novel fundamental property that we name mode rejection. As illustrated in Fig. 1c, d, this effect can be seen as the inverse of the attraction process. It is only by moving to waveguides supporting $N > 2$ modes that mode rejection can be properly understood and distinguished from mode attraction.

In this work, the waveguides of choice are multimode and multi-core optical fibres. It should be noted however that our results can be promptly adapted to different optical (e.g. integrated waveguides) or even non-optical systems exhibiting Kerr-like nonlinearity. The choice of multimode/multicore fibres is driven by two considerations. Firstly, these fibres benefit from a decade of improved manufacturing processes as well as a wide array of devices for efficient mode manipulation, decomposition and characterisation, which are key steps in our investigation. Secondly, these fibres represent versatile platforms for exploring not only fundamental physics but also new applications. Here, we provide an example of coherent beam combination based on the new notion of mode rejection.

Our outcomes broaden our knowledge on the complex nonlinear multimode dynamics and open the path towards a more general idea of all-optical mode control. More generally, they shed new light into the fundamental ideas of attraction, rejection and control in the framework of nonlinear multimode systems, including but not limited to optical fibres.

## Results

### Theory of mode attraction and rejection

Our theoretical investigation focuses on two distinct scenarios. Firstly, we consider a polarisation maintaining (PM) fibre of length $L$, supporting the propagation of an arbitrary number $N$ of spatial modes linearly polarised along one of its birefringence axes. The FS (BCB) enters the fibre at $z = 0$ ($z = L$) and excites a combination of spatial modes, whose amplitude is indicated by $f_n(z,t)$ ($b_n(z,t)$), $n = \{1, 2, \ldots N\}$.

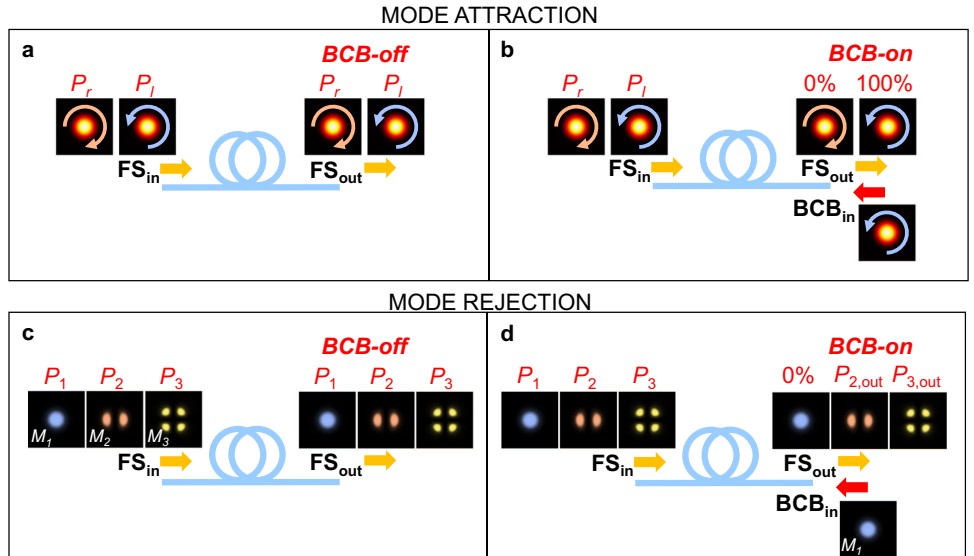

**Fig. 1 | Illustration of mode attraction and rejection in multimode waveguides.** The input FS ($FS_{in}$) and input BCB ($BCB_{in}$) are launched at the two opposite ends of an optical fibre. **a, b** Mode attraction in single-mode isotropic optical fibres. In these fibres, the left and right circular polarisation states play the role of robust attractors. When the BCB is turned off (**a**), the mode content of the output FS ($FS_{out}$) mirrors the input one (right and left-circular polarisations have respectively relative powers $P_r$ and $P_l$, where $P_r$ and $P_l$ are arbitrary and $P_r + P_l = 100\%$). When the BCB is turned on and has a similar power to the FS (**b**), their nonlinear interaction modifies the mode content. If the input BCB is coupled to one circular polarisation mode (left circularly polarised in this example), then irrespective of the mode content of the input FS, the output FS is attracted to the mode of the input BCB (100% of the

power in the left circularly polarised mode). **c, d** Mode rejection in multimode optical fibre. In this example the fibre supports 3 modes indicated with $M_1, M_2, M_3$. When the BCB is turned off (**c**), the mode content of the output FS mirrors the input one (relative power $P_1$ for mode $M_1$, $P_2$ for $M_2$ and $P_3$ for $M_3$, where $P_1$, $P_2$ and $P_3$ are arbitrary and $P_1 + P_2 + P_3 = 100\%$). When the BCB is turned on and has a similar power to the FS (**d**), their nonlinear interaction modifies the mode content. If the input BCB is fully coupled to one spatial mode ($M_1$ in this example), then irrespective of the mode content of the input FS, the output FS rejects the mode of the input BCB (0% power in mode $M_1$). The specific amount of power on the remaining modes ($P_{2,out}$ for $M_2$ and $P_{3,out}$ for $M_3$, where $P_{2,out} + P_{3,out} = 100\%$) depends on the system parameters.

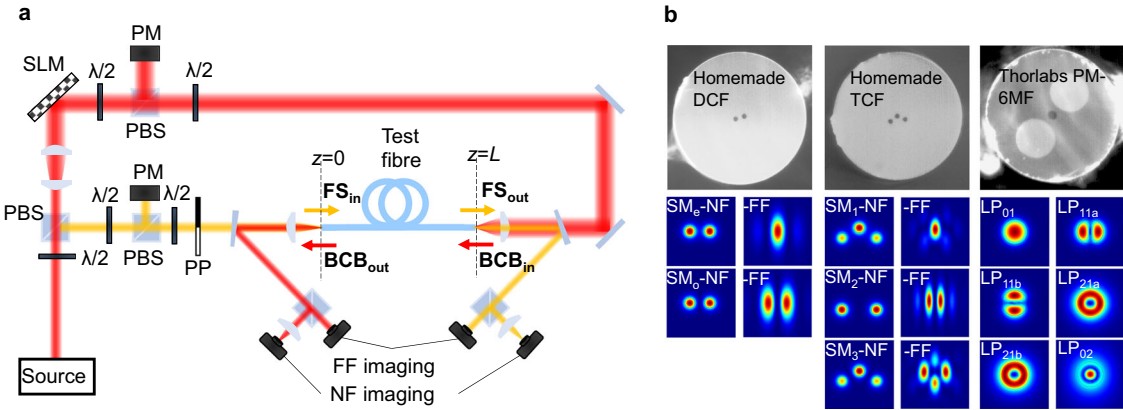

**Fig. 2 | Experimental setup and test fibres. a** Schematic of the experimental setup and beam path for the FS (in yellow) and the BCB (in red). PBS=polarisation beam splitter; SLM=spatial light modulator; PM=power meter; PP=phase plate; FF=far-field; NF=near-field. The fibre ends $z = 0$ (input FS, output BCB) and $z = L$ (output FS, input BCB) are indicated with black dashed lines. **b** Microscope image of the 3 test fibres and related spatial modes used in the experiments: a homemade dual-core fibre (DCF) supporting 2 modes ($SM_{e,o}$), a homemade tri-core fibre (TCF) supporting 3 modes ($SM_{1,2,3}$) and a commercial polarisation maintaining fibre (PM-6MF)

from Thorlabs supporting 6 distinct modes at $\lambda = 1040$ nm ($LP_{01,11a,11b, 21a,21b,02}$). In our experiments each spatial mode is linearly polarised along one of the birefringence axes of the fibres under test (see Supplementary information note 3). The substantial birefringence of the fibres ensures that the input polarisation is maintained ($> 10$ dB polarisation extinction ratio). The spatial mode profiles in the near-field (NF) and the far-field (FF) of the DCF and TCF are reported at the bottom and can be compared against the camera images shown in Figs. 3–5 to appreciate the mode rejection dynamics. For the Thorlabs fibre just the NF modes are reported.

We find that their spatio-temporal evolution is described by the set of coupled nonlinear Schrödinger equations Eq. (1) (see Methods).

Secondly, we consider the case of a single-mode isotropic fibre. Differently from the previous case, it can only support the propagation of $N = 2$ modes. These are quasi-degenerate modes with orthogonal circular polarisation, whose dynamics is well-known and is described by Eq. (2) with $N = 2$ (see Methods). One of the key-points of our analysis is that we extend the study of Eq. (2) to an arbitrary dimension $N > 2$. Despite the existence of fibre systems where Eq. (2) with $N > 2$ could apply is yet unknown, however this extension allows for a direct comparison between Eq. (1) and Eq. (2) in the general case $N > 2$, which ultimately leads to a new understanding of the nonlinear mode dynamics in counter-propagating systems.

We note that Eq. (1) and Eq. (2) are similar but differ in the last term on the right-hand side, which describes the power exchange among forward and backward modes. The main outcome of our theoretical investigation is that, under certain conditions on the Kerr coefficients, both Eq. (1) and Eq. (2) are integrable: therefore, they can be solved analytically (see Methods and Supplementary information note 1). The analytical solution discloses a new fundamental property.

In the case of Eq. (1), by calling $P_f = \sum_n |f_n|^2$ ($P_b = \sum_n |b_n|^2$) the total injected FS (BCB) power and $D_R(z)$ the normalised correlation coefficient $D_R(z) = \sum_n f_n(z) b_n(z) / Q$ with $Q = (P_f P_b)^{1/2}$, one finds that $D_R(L)$ decreases as $1/(L \cdot Q)$: therefore, $|D_R(L)| \rightarrow 0$ for sufficiently large $L$ or $Q$. Mode rejection is a direct consequence of this property. Indeed, if the input BCB is launched into an individual mode only, say the mode-$m$, such that $|b_m(L)| = P_b^{1/2}$ and $b_n(L) = 0 \; \forall \; n \neq m$, then $|D_R(L)|$ reduces to $|f_m(L)|/P_b^{1/2}$. The condition $|D_R(L)| \rightarrow 0$ implies therefore that $f_m(L) \sim 0$: namely, irrespective of the mode distribution of the input FS in $z = 0$, the output FS in $z = L$ carries no energy in the mode-$m$, that is to say the mode-$m$ is rejected at the output.

On the other hand, Eq. (2) are characterised by opposite dynamics. Indeed, if we call $D_A(z) = \sum_n f_n(z) b_n^*(z) / Q$, then $|D_A(L)| \rightarrow 1$ for increasingly large $L$ or $Q$. Now, if the input BCB is launched into an individual mode only, say, the mode-$m$, then $|D_A(L)| = |f_m(L)|/P_f^{1/2} \rightarrow 1$ and therefore $|f_m(L)|^2 \sim P_f$, which means that all the power of the output FS gets coupled into the same mode-$m$ of the input BCB. This generalises the idea of mode attraction in the case of $N$-dimensional systems, with $N$ arbitrary.

In conclusion, Eqs. (1) and (2) describe the two opposite processes of mode rejection and attraction in the case of arbitrary dimension $N$.

Their difference is intimately related to the nature of the energy-exchange term that couples the forward and the backward beams. The schematic in the Supplementary information note 5 summarises these results. We highlight that it is only in systems supporting more than 2 modes ($N > 2$) that one can fully appreciate the idea of mode rejection. Indeed, in bimodal systems the rejection of one mode could be misleadingly interpreted as the attraction towards the other mode[34,39].

Our theoretical analysis is supported by a simulation tool that numerically solves Eqs. (1) and (2). Besides confirming the theoretical predictions, the simulations reveal that mode rejection and attraction take place in the most general case where the Kerr coefficients are fully arbitrary (See Supplementary information note 2). Therefore, these phenomena represent universal features of nonlinear counter-propagating systems.

## Experimental results on mode rejection

As anticipated, the existence of optical fibres where Eq. (2) with $N > 2$ may apply is yet unknown. On the other hand, the case of Eq. (1) with $N > 2$ is relevant to PM multimode fibres. Therefore, in our experiments we have conducted a systematic investigation in a variety of PM multimode and multicore fibres, in order to provide the first experimental evidence of the mode rejection principle. Details on the fibres and experimental parameters are provided in Fig. 2b and Supplementary information note 3, including the choice of the optical power to achieve the high levels of system nonlinearity required. In our setup (Fig. 2a), an in-house built linearly polarised source (central wavelength $\lambda = 1040$ nm, 500 ps pulses) is split into two beams that form the input FS and BCB. Cameras are placed in the near- and far-field to measure the spatial intensity profiles of the output beams, and these images are then used to estimate the mode content of the output FS via a mode decomposition technique covering all guided modes of the fibre (see Methods and Supplementary information note 4 for details).

The first fibres under test were a homemade dual-core fibre (DCF) and tri-core fibre (TCF). Multicore fibres represent a novel platform for exploring nonlinear effects in counter-propagating systems, which allow generalisation of our theoretical predictions beyond standard single-core multimode fibres. Furthermore, these fibres allow us to characterise the robustness of mode rejection against a variety of simultaneous fabrication imperfections, which include random variations of core radii, shape and core-to-core distance. Spatial light modulators and phase-plates were used to control the intensity and

phase of light coupled into each core, and hence to excite an arbitrary combination of modes.

Figure 3 displays the experimental results obtained in a 1-m long DCF that supports the propagation of an even and an odd supermode, here indicated with $SM_e$ and $SM_o$. The FS and BCB are co-polarised. On one side, the input FS is launched with fixed power, coupled to different arbitrary combinations of modes. At the opposite fibre end, the input BCB is launched with variable power, and it is coupled to the odd (Fig. 3a) or the even (Fig. 3b, c) supermode. In line with our theoretical predictions, experiments confirm that, for increasing values of BCB power, the output FS gradually rejects the BCB mode. The numerical simulation of Eq. (1) shows a good match with the experiments when using comparable parameters (pulse width, peak powers and Kerr coefficients, see also Supplementary information note 3).

Note that, irrespective of the mode distribution of the input FS, according to our theoretical model and our numerical simulations full rejection (i.e. no output FS power coupled into the input BCB mode) could be in principle achieved by appropriately increasing the forward and backward powers $P_f$ and $P_b$. However, in our experiments the maximum coupled peak power $P_f + P_b$ is limited to 12 kW by the optical source in use, which explains why only partial rejection was observed. It is worth noting that the power requirement could be substantially mitigated by resorting to highly nonlinear fibres, which includes fibres with non-silica host.

As previously mentioned, when a fibre supports just two guided modes, as is the case for the DCF, then rejection of one mode corresponds to attraction towards the other mode. This opens up the possibility to lock the output FS to either the even or the odd supermode in an all-optical way. In the case under analysis, when the BCB is coupled to the odd (even) supermode, then the output FS undergoes rejection of that mode, and consequently it is attracted towards the even (odd) supermode, which corresponds to a condition of in-phase (out-of-phase) coherent combination of the two cores. This is confirmed by the observation of the far-field intensity profile of the output FS on the camera, which shows a transition towards an in-phase (out-of-phase) coherent combination when increasing the power of the BCB coupled to the odd (even) supermode, as reported in Fig.3 and

in the Supplementary movies 1–3 (see Supplementary information note 4).

A striking property of mode rejection is its robustness against external perturbations, which is illustrated in Fig. 4a. Moving and/or bending the fibre modifies the launching conditions of the FS and may lead to random coupling among the fibre modes. When the BCB is turned off, this results in random variations of the output FS. However, when the BCB is turned on and is coupled to one mode ($SM_e$ in Fig. 4a), then mode rejection takes place, irrespective of the launching conditions. As a result, we observe the effective rejection of that mode in the output FS (relative power of $SM_e < 10\%$ in Fig. 4a). Consequently, we can robustly lock the output FS to either one or the other supermode.

A further relevant feature of mode rejection is that it can be controlled via the relative angle of polarisation orientation between the FS and the BCB. When the FS and BCB are orthogonally polarised, the power-exchange term of Eq. (1) is reduced by a factor of 3 (see Methods), which leads to a substantial suppression of the power exchange between the FS and BCB, ultimately compromising the mode rejection process. This is confirmed by the results in Fig. 4b, which shows the evolution of mode rejection as a function of the relative angle $\alpha$ between the input FS and the BCB in the DCF. When $\alpha = 0$ deg (i.e. the FS and the BCB are co-polarised), a strong rejection of the BCB mode is observed (relative power of $SM_e = 10\%$ in Fig. 4b), which gradually reduces for increasing values of $\alpha$. When $\alpha = 90$ deg (i.e. the FS and BCB are orthogonally-polarised), the far-field intensity of the output FS resembles the case where the BCB is turned off, indicating that mode rejection is substantially suppressed.

Our experiments with the TCF, reported in Fig. 5, show similar outcomes, and again suggest that mode rejection is robust against standard fibre fabrication imperfections (see Supplementary information note 3). Indeed, the TCF supports 3 guided supermodes ($SM_1, SM_2, SM_3$), and in good agreement with the numerical simulations mode rejection is observed when the BCB is coupled to any of these.

Further experiments were carried out in a single-core few-mode PM fibre supporting up to 6 modes at $\lambda = 1040$ nm. Although the specific evolution of the power carried by each mode is dependent on the input FS mode content, rejection of the input BCB mode was consistently achieved, irrespective of the input FS. Figure 6 reports a

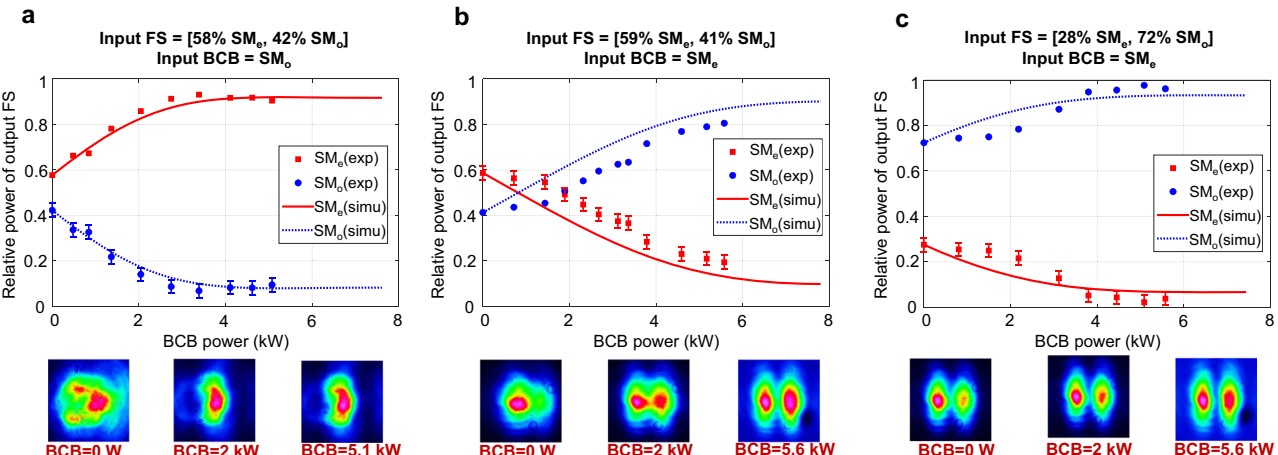

**Fig. 3 | Mode rejection in the DCF.** Mode decomposition of output FS in the DCF fibre (1 m long) for different launching conditions (experiments(exp): dots; simulations(simu): lines). The bottom images show the far-field intensities of the output FS for 3 distinct values of BCB power. The input FS is coupled to different combinations of $SM_e$ and $SM_o$ (see relative power of input FS on the top of each panel) and launched with fixed power of 3.75 kW, 6.5 kW and 6.2 kW in (**a**–**c**) respectively. If the input BCB is coupled to $SM_o$ (**a**), then the output FS rejects $SM_o$ and is therefore mainly coupled to $SM_e$. When the BCB power = 5.1 kW, ~90% of the output FS power is coupled to $SM_e$. Consequently, the output FS far-field exhibits one single lobe and resembles the mode $SM_e$-FF of Fig. 2b, corresponding to in-phase combination of the cores. On the contrary, if the input BCB is coupled to $SM_e$ (**b**, **c**), then the output FS undergoes rejection of $SM_e$ and is therefore mainly coupled to $SM_o$. When the BCB power = 5.6 kW, ~81% and 98% of the output FS power is coupled to $SM_o$ in cases **b** and **c**, respectively. Consequently, the output FS far-field exhibits two distinct symmetric lobes and resembles the mode $SM_o$-FF of Fig. 2b, corresponding to out-of-phase combination of the cores. Error bars of ±3% are added to the measured relative power of the rejected mode, which represents the estimated uncertainty of our mode decomposition algorithm.

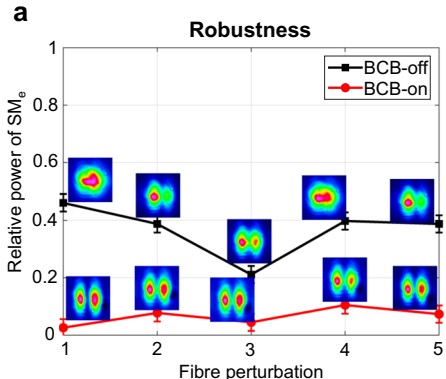

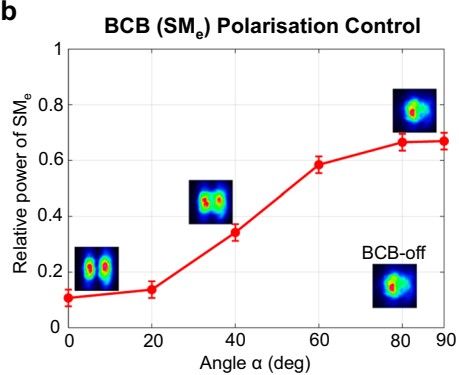

**Fig. 4 | Robustness and control of mode rejection.** Input FS and BCB are linearly polarised and launched with ~ 5 kW peak power in the DCF. **a** Mode decomposition of the output FS for off/on BCB. The input FS is coupled to a combination of the two supermodes, whereas the input BCB is coupled to $SM_e$. FS and BCB are co-polarised. The fibre is perturbed 5 different times. Each time, the perturbation consists in compressing or bending the fibre with different levels of intensity and in different points. For each perturbation, the mode decomposition of the output FS is computed, and the output FS far-field intensity is imaged when the BCB is either turned off or on. When BCB is on, robust rejection of $SM_e$ occurs in the output FS (< 10% power in all 5 cases). **b** Mode decomposition and far-field intensity of the output FS as a function of the relative angle $\alpha$ between the polarisation orientations of the input FS and the BCB. The far-field with BCB off is reported for comparison. Error bars of ±3% are added to the measured relative power, which represents the estimated uncertainty of our mode decomposition algorithm.

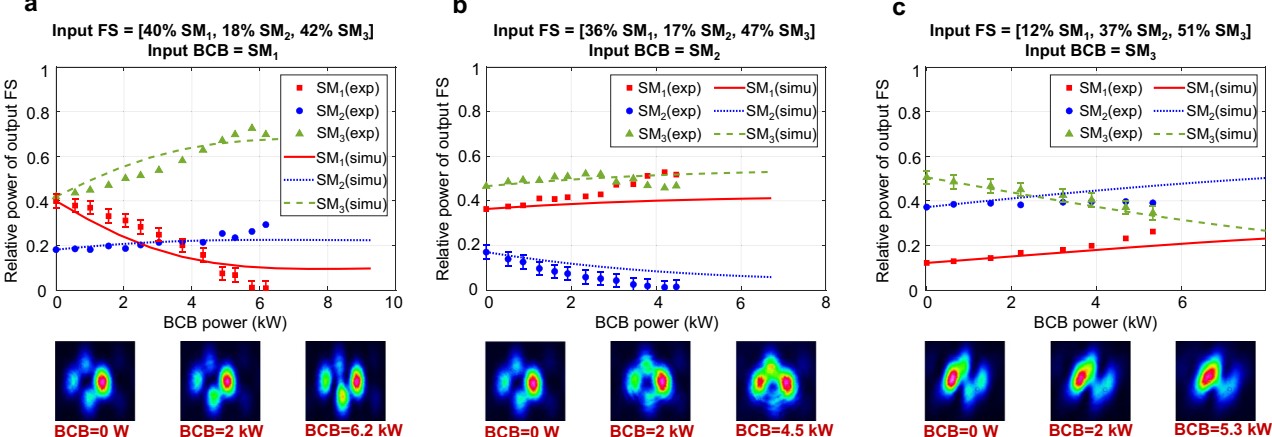

**Fig. 5 | Mode rejection in the TCF.** Mode decomposition of the output FS in the homemade TCF (40 cm long) for 3 different launching conditions, and comparison with numerical simulations. The input FS is coupled to different combinations of guided supermodes $SM_{1,2,3}$ (see relative power of input FS on the top of each panel) and launched with fixed power of 4.23 kW, 4.23 kW and 4.33 kW in (**a**–**c**) respectively. The input BCB is mainly coupled to $SM_1$(**a**), $SM_2$(**b**) or $SM_3$(**c**). We observe almost full rejection of $SM_1$ and $SM_2$ in (**a** and **b**), respectively, when the BCB has ~ 6.2 kW(**a**) or 4.5 kW(**b**) peak power. Rejection of $SM_3$ reported in **c** is instead less effective. Nevertheless, we observe a clear trend of rejection in line with numerical simulations, indicating that full rejection maybe achieved by increasing the total power of FS and/or BCB. Error bars of ±3% are added to the measured relative power of the rejected mode, which represents the estimated uncertainty of our mode decomposition algorithm.

few illustrative examples (see also Supplementary information note 4 and related Supplementary movies 7–9).

### Dynamic of the backward control beam

Equation (1) are invariant with respect to the exchange between FS and BCB modes. However, different boundary conditions apply to the input FS and to the input BCB, which reflects their different roles.

Because the input BCB plays the role of the control beam, its mode content is fixed and coupled to one single mode. For this reason, the output FS systematically undergoes rejection irrespectively of the input FS mode content. On the other hand, the input FS plays the role of a probe beam with arbitrary mode content. Therefore, the output BCB does not undergo any rejection dynamics, because different input FSs lead to different output BCB mode contents.

It is interesting to note that, in the special case of a two-mode-fibre having the Kerr coefficients all identical, the mode content of the output BCB is attracted towards the orthogonal modal state of the input FS[34] when they have the same power. This is confirmed by our experiments in the home-made DCF, where the above-mentioned condition on the Kerr coefficients is met with excellent approximation ($\gamma_{11} \sim \gamma_{12} \sim \gamma_{22}$, see Supplementary information note 3). In the example illustrated in Fig. 7 the input BCB is coupled to supermode $SM_o$, whereas the input FS to a combination of 60% $SM_e$ and 40% $SM_o$. As expected, the output FS undergoes rejection of the input BCB mode. On the other hand, the output BCB achieves a mode content of ~40% $SM_e$ and ~60% $SM_o$ when the two beams have the same power (see green dashed vertical line).

These considerations have an important consequence when the input FS has a random mode content in time. In that case the amount

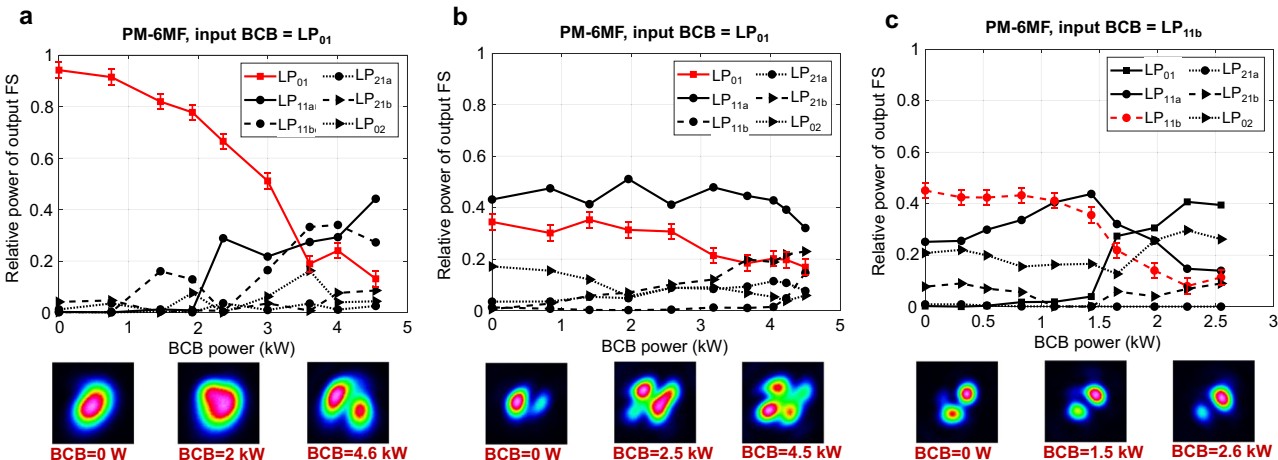

**Fig. 6 | Mode rejection in the 6-mode fibre (PM-2000 from Thorlabs, 1 m long).** The input FS is coupled to different combinations of modes and launched with fixed power of 4.00 kW, 4.10 kW and 3.20 kW in (**a**–**c**) respectively. The input BCB is mainly coupled either to $LP_{01}$ (**a**, **b**) or $LP_{11b}$ (**c**). Error bars of ±3% are added to the measured relative power of the rejected mode, which represents the estimated uncertainty of our mode decomposition algorithm.

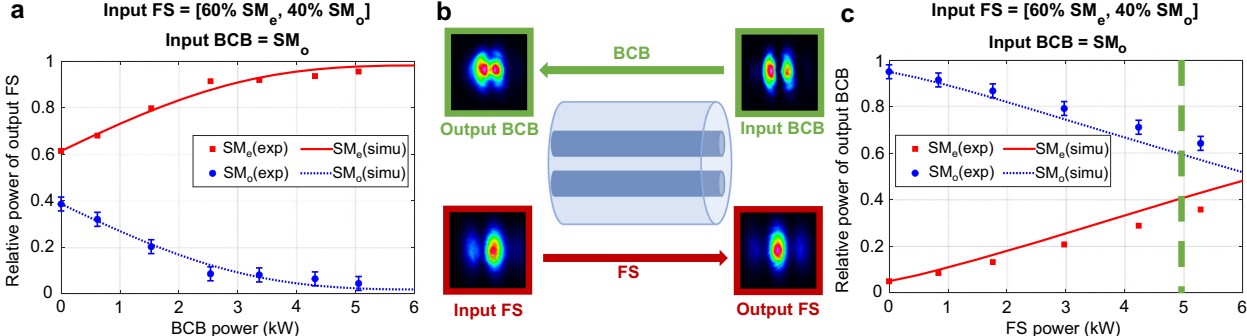

**Fig. 7 | Comparison between the dynamics of the output FS and the output BCB in the home-made DCF.** In the case of panel **a**, the input FS is launched with fixed power of 5.05 kW, whereas the BCB power is variable (see horizontal axis). In the case of (**c**), the input BCB is launched with fixed power of 5.05 kW, whereas the FS power is variable (see horizontal axis). **b** Shows the far-field intensities. The input FS is a combination of ~60% $SM_e$ and ~40% $SM_o$, whereas the input BCB is coupled to $SM_o$. Consequently, the output FS undergoes rejection of $SM_o$ as illustrated in (**a**).

On the other hand, the output BCB does not undergo rejection, as illustrated in (**c**). Indeed, the mode content of the output BCB in stationary regime strictly depends on the fibre parameters and the input FS, which is in general arbitrary. In this example, because the fibre is bimodal and the Kerr coefficients are almost identical, then the output BC reaches a state orthogonal to the input FS ( ~ 40% $SM_e$ and ~60% $SM_o$) when the 2 beams have the same power (5.05 kW, see green dashed line in **c**).

of disorder of the input FS is transferred to the output BCB, whose mode content becomes therefore randomly distributed. This dynamic is similar to that of polarisation attraction phenomena in single-mode fibres[34], where a mutual exchange in the degree of polarisation is achieved between FS and BCB, that is to say, the polarisation attraction undergone by the output FS takes place at the expense of a depolarisation of the output BCB.

**Towards the idea of mode control**

Mode attraction and rejection take place when the input BCB is coupled to one individual mode. We envisage further novel scenarios in the most general case where the input BCB is coupled to an arbitrary set of modes. In this framework, one may wonder whether for a given input FS it is possible to achieve a specific output FS mode distribution by opportunely setting the input BCB. Once again the theoretical framework developed for the special case $\gamma_{nn} = \gamma$ and $\gamma_{nm} = (1/2)\gamma$ $(n \neq m)$ gives a clue on some possible scenario. For example, following the steps introduced in the Supplementary information note 1, we find that for a given input FS it is possible to focus all the output FS power into one single mode, say mode-$k$, provided that the input BCB has the

following configuration: $|\hat{b}_k(L)| \sim 0$, $\hat{b}_n(L) \sim -i|\hat{f}_n(0)|e^{i\phi_{kn}}/r$ if $n \neq k$, where $r = (1 - |\hat{f}_k(0)|^2)^{1/2}$ and $\phi_{kn}$ is the relative phase between the input FS mode-$k$ and mode-$n$.

Note that this outcome should not be confused with the notion of mode attraction, which is fundamentally different. Indeed, mode attraction implies that for a fixed input BCB coupled to one mode, then the output FS is attracted towards that mode for any arbitrary input FS. On the contrary, in the case mentioned above, the input BCB is not fixed, but depends on the specific input FS mode power distribution and relative phase.

We have performed some preliminary experiments in our TCF where the input BCB is coupled to a combination of modes, rather than just one mode. In Fig. 8 we observe that, for the same input FS, different input BCBs give rise to substantially different output FS mode distributions.

These preliminary results pave the way towards a more general idea of mode control, that is to say, the ability to control on demand and all-optically (and then potentially at ultrafast rate) the mode content of the output FS via the BCB. The latter may be even

**a**

Input FS = [20% SM$_1$, 59% SM$_2$, 21% SM$_3$]
Input BCB = [79% SM$_1$, 2% SM$_2$, 19% SM$_3$]

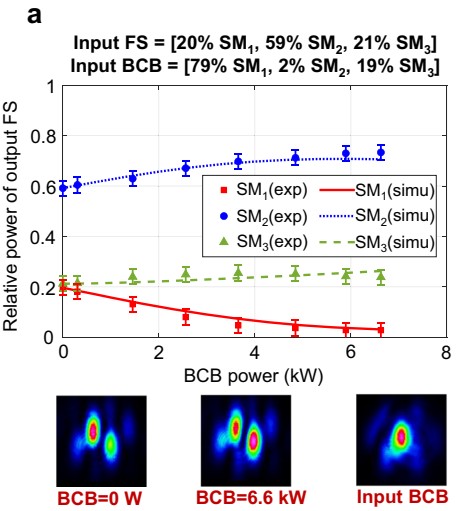

**b**

Input FS = [16% SM$_1$, 61% SM$_2$, 23% SM$_3$]
Input BCB = [20% SM$_1$, 28% SM$_2$, 52% SM$_3$]

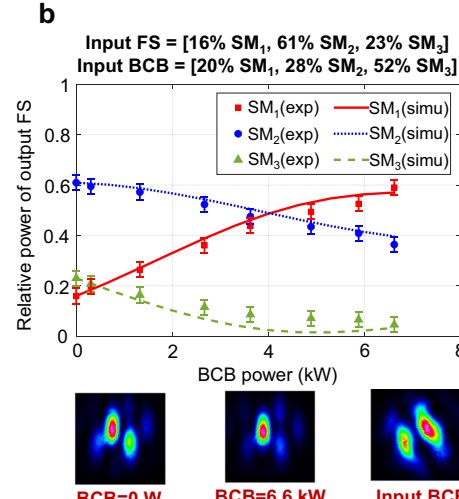

**Fig. 8 | Preliminary experiments of all-optical mode control.** Differently from Figs. 3–7, where the input BCB is coupled to one single mode leading to mode rejection, now the input BCB is coupled to a combination of modes (see relative power of input FS and BCB on the top of **a** and **b**). The bottom images show the far-field intensities of the output FS when the BCB power is either 0 W or 6.6 kW. The input FS is almost identical in (**a** and **b**), and consequently the corresponding output FS is almost identical in the two cases when BCB is turned off (see the far-field with BCB = 0 W in **a** and **b**). On the contrary, the input BCB is different in (**a** and **b**). We see that different mode distributions of the input BCB give rise to different mode distributions of the output FS (see the far-field with BCB = 6.6 KW in **a** and **b**). These results seem to suggest that, by properly setting the input BCB, the output FS could be shaped on demand in time and space. Error bars of ±3% are added to the measured relative powers, which represents the estimated uncertainty of our mode decomposition algorithm.

generated via back-reflection of the FS, thus without the need for an independent and additional optical source[31]. A new idea of light-self organization then emerges, where the output FS is shaped in time and space depending on its input state, leading to new opportunities like all-optical switching or highly scalable coherent combination in multimode and multicore waveguides, to name a few. These ideas are of course speculative at the moment and additional extensive investigation will be required to put them into practice. However, the theoretical and experimental results reported in this work represent ideal tools to address the above-mentioned opportunities and related challenges.

## Discussion

In this work, we investigated the nonlinear dynamics of multimode optical waveguides with a counter-propagating geometry. We identified two classes of multimode systems that exhibit the two opposite dynamics of mode rejection and attraction.

The first class is characterised by the occurrence of mode rejection, and is described by Eq. (1). It is only when moving to systems supporting more than two modes that the peculiar dynamics of mode rejection can be properly understood and differentiated from mode attraction. We provide systematic evidence of mode rejection in a variety of home-made multicore fibres and commercial multimode fibres. Our experimental outcomes highlight the robustness of mode rejection against standard fabrication imperfections and external perturbations, as well as the ability to control this process via the relative state of polarisation among the FS and the BCB.

The second class of multimode systems is characterised by effective mode attraction towards the BCB mode, and is described by Eq. (2). Isotropic single-mode fibres exhibiting polarisation mode attraction represent the lowest-dimensional example ($N = 2$) belonging to this class. Although polarisation mode attraction has been predicted and demonstrated almost two decades ago, our theoretical analysis generalises the idea of mode attraction to systems of arbitrary dimension $N$. The scope of this generalisation is two-fold. Firstly, it allows a direct comparison between

Eqs. (1) and (2), and therefore between the fundamental notions of rejection and attraction, in systems of arbitrary dimension $N$. Second, it sheds light on the universality of these processes, which goes beyond bidimensional systems. Although currently it is yet unknown the existence of multimode systems where Eq. (2) with $N > 2$ applies, however these may be found in the future, either in optical fibres or different physical systems.

We note that the ideas of mode attraction and rejection are intrinsically related to an input BCB that is coupled to one single mode. However, an even more rich scenario emerges when the input BCB is coupled to a combination of modes, which brings forward the more general idea of modal control. This may inspire a plethora of new applications, including all-optical switching for space-division-multiplexing, or tuneable coherent combination for high-power lasers, of which the experimental outcomes illustrated in Figs. 3 and 4 are indeed a basic example in the case of a dual-core fibre.

Our theoretical analysis may be adapted to the case where the BCB (FS) is generated as a back-reflection of the FS (BCB) through a set of mirrors[31] or a cavity. This could lead to new understanding and concepts in multimode optical parametric oscillators and lasers, and related applications.

In conclusion, by predicting the existence of mode attraction and rejection in multimode waveguides of arbitrary dimension $N$, and by providing experimental evidence of the latter, this work sets the basis for a novel understanding of the complex dynamics in nonlinear multimode waveguides and suggests new scenarios and applications that can be explored by means of the theoretical, numerical and experimental tools introduced in this paper. The schematic in the Supplementary information note 5 reports a conceptual map of the above-mentioned outcomes.

It is worth noting that the ideas and theoretical results introduced in this paper encompass but are not limited to optical waveguides. The fundamental notions of attraction and rejection may be used to investigate any nonlinear system described by formally similar sets of equations, which includes classical wave thermalisation and condensation phenomena[40–42] or non-optical systems like hydrodynamics[43,44] and matter-wave Bose-Einstein condensates[45].

# Methods

## Theoretical framework and numerical simulations

The following theoretical analysis applies to multimode and multi-core fibres that, as previously mentioned, represent the waveguides of choice in our experiments. Nevertheless, this analysis could be readily adapted to different kind of Kerr-nonlinear waveguides by taking into account the related specific nonlinear cubic response. Here we consider two beams centred at the same carrier wavelength that are counter-propagating in a multimode optical fibre and coupled to a combination of $N$ guided spatial modes. We focus on a polarisation-maintaining (PM) step-index geometry as it represents the ideal scenario to investigate the complex multimode dynamics, without the burden induced by random polarisation variations.

We follow a standard procedure starting from the Maxwell equations and then we ignore the fast-oscillating terms that average out to $0$[34,46,47]. We also ignore Raman scattering, which indeed turns out to be negligible for the power levels used in our experiments. We finally derive the following set of coupled Schrödinger equations for the forward and backward modal amplitudes:

$$\partial_z f_n + v_n^{-1}\partial_t f_n = -\,\mathrm{i}\gamma_{nn}|f_n|^2 f_n + \mathrm{i}f_n \sum_{\substack{m=1}}^{N}\gamma_{nm}\left(\kappa|b_m|^2 + 2|f_m|^2\right)$$

$$+\,\mathrm{i}\kappa b_n^* \sum_{\substack{m=1\\m\neq n}}^{N}\gamma_{nm}b_m f_m$$

$$-\partial_z b_n + v_n^{-1}\partial_t b_n = -\,\mathrm{i}\gamma_{nn}|b_n|^2 b_n + \mathrm{i}b_n \sum_{\substack{m=1}}^{N}\gamma_{nm}\left(\kappa|f_m|^2 + 2|b_m|^2\right)$$

$$+\,\mathrm{i}\kappa f_n^* \sum_{\substack{m=1\\m\neq n}}^{N}\gamma_{nm}b_m f_m \tag{1}$$

Where: $\kappa=2$ for linearly co-polarised counter-propagating beams, whereas $\kappa=2/3$ when these are orthogonally polarised; $f_n(z,t)$ and $b_n(z,t)$ are respectively the amplitude of the forward and backward mode-$n$; $v_n$ is the related group velocity; $\gamma_{nm}$ are the nonlinear Kerr coefficients computed from the intramodal and intermodal effective areas[48] (see Supplementary information note 3 for details on the parameters). The latter require the calculation of the spatial mode profiles, which are computed with a finite-element-method software (Comsol Multiphysics 5.6) and take into account the actual refractive index profile of the fibres under test as measured by means of an Optical Fiber Analyser (Rayphotonics IFA-100). While the first and second term on the right-hand-side of Eq. (1) describe respectively self and cross-phase modulation, the last term represents the intermodal power exchange among forward and backward modes. In order to simplify our analysis, the group velocity dispersion (GVD) and the higher-order dispersion terms are not included in Eq. (1)[34,49]. Indeed, the largest GVD coefficient at the wavelength of interest ($\lambda = 1040$ nm) is $\beta_{2\mathrm{max}} \sim 25$ ps$^2$/km, whereas the pulses used in our experiments are $t_0 = 0.5$ ns wide. This corresponds to a minimum dispersion length $t_0^2/\beta_{2\mathrm{max}}$ of several km, therefore order of magnitudes longer than the fibres under test. Similarly, propagation losses are negligible due to the short length of the fibres in use.

Note that in the counter-propagating setup the input forward and backward fields are injected respectively at $z = 0$ and $z = L$. Therefore, the boundary conditions for Eq. (1) that fix the initial state of the forward and backward modes are $f_n(z = 0,t)$ and $b_n(z = L,t)$, respectively. A further boundary condition defines the fields at $t = 0$ inside the fiber ($0 < z < L$). We assume the fibre is empty, that is to say, $f_n(z,t = 0) = b_n(z,t = 0) = 0$.

If one considers now two counter-propagating beams in an isotropic single-mode fibre, their spatio-temporal dynamics is described by the following set of equations[49] with $N = 2$ and $\kappa=2$:

$$\partial_z f_n + v_n^{-1}\partial_t f_n = -\,\mathrm{i}\gamma_{nn}|f_n|^2 f_n + \mathrm{i}f_n \sum_{\substack{m=1}}^{N}\gamma_{nm}\left(\kappa|b_m|^2 + 2|f_m|^2\right)$$

$$+\,\mathrm{i}\kappa b_n \sum_{\substack{m=1\\m\neq n}}^{N}\gamma_{nm}b_m^* f_m$$

$$-\partial_z b_n + v_n^{-1}\partial_t b_n = -\,\mathrm{i}\gamma_{nn}|b_n|^2 b_n + \mathrm{i}b_n \sum_{\substack{m=1}}^{N}\gamma_{nm}\left(\kappa|f_m|^2 + 2|b_m|^2\right)$$

$$+\,\mathrm{i}\kappa f_n \sum_{\substack{m=1\\m\neq n}}^{N}\gamma_{nm}b_m f_m^* \tag{2}$$

Where $f_1$ and $f_2$ ($b_1$ and $b_2$) indicate the modal amplitude of the forward (backward) right and left-circular polarisation modes, respectively. However, as anticipated in the section Results, in this work we extend the study of Eq. (2) to the most general case of arbitrary dimension $N > 2$.

The difference between the intermodal power exchange terms in Eq. (1) and Eq. (2) is ultimately responsible for drastically different dynamics, which results in mode rejection in the case of Eq. (1), whereas attraction in Eq. (2).

Equations (1) and (2) have been solved via a standard finite-difference method[49]. The validity and the robustness of our numerical algorithm have been tested both via comparison with a different integration method (shooting method) as well as direct comparison with analytical (see Supplementary information note 2) and experimental (see Figs. 3–5) results.

Similarly to what has been reported in previous studies of polarisation attraction in single-mode fibres[33], our numerical simulations show that when the input mode amplitudes are steady ($f_n(0,t) \equiv f_n(0)$, $b_n(L,t) \equiv b_n(L)$), then after a transient time the fields relax towards a stationary state $f_n(z) \equiv f_n(z,t)$ and $b_n(z) \equiv b_n(z,t)$ at any point inside the fibre (see Supplementary Figs. 2–3). Note that in the experiments, rather than steady input fields, ns pulses are used to enhance the peak power and hence the system nonlinearity. However, this does not change the rejection and attraction dynamics under investigation, as far as one replaces the fibre length with the actual interaction length $L_{in} = 2 \cdot t_0 \cdot c$ ($c$=speed of light in the fibre) among the forward and backward pulses, and once the modal amplitudes are replaced with their own average value over the pulse duration.

We note that if FS and BCB are centred at different carrier wavelengths, say $\lambda_F$ and $\lambda_B$ respectively, then the intermodal power exchange interactions (last terms on the right-hand-side of Eqs. (1) and (2)) are subject to a nonlinear phase mismatch $\Delta\beta_{nm} = \beta_n(\lambda_F) - \beta_n(\lambda_B) + \beta_m(\lambda_B) - \beta_m(\lambda_F)$, where $\beta_n(\lambda)$ is the propagation constant of the mode-$n$ at wavelength $\lambda$. The overall dynamics discussed in our manuscript remain unchanged whenever the interaction length $L_{in}$ is much shorter than the corresponding beat length $2\pi/\Delta\beta_{nm}$. By expanding in Taylor series the phase-mismatch $\Delta\beta_{nm}$, we find after some algebra that the condition $L_{in} \ll 2\pi/\Delta\beta_{nm}$ can be recast as follows: $\Delta\lambda \ll \lambda_0^2/(c\, L_{in}\,\Delta\beta_{1,nm})$, where $\Delta\lambda = \lambda_F - \lambda_B$, $\lambda_0 = (\lambda_F + \lambda_B)/2$ is the central wavelength and $\Delta\beta_{1,nm}$ is the differential inverse group velocity between mode-$n$ and mode-$m$. In the case of the fibres under test, whose parameters are reported in Supplementary information note 3, we find $\Delta\lambda \ll 25$ nm, therefore the system dynamics would be unaffected if FS and BCB are detuned a few nanometres apart.

## Theory of mode rejection

As previously mentioned, after a transient time the mode amplitudes typically achieve stationary states $f_n(z) \equiv f_n(z,t)$ and $b_n(z) \equiv b_n(z,t)$.

These are used in the subsequent analysis. Along with the correlation coefficient $D_R(L) = \sum f_n(L)b_n(L)/Q$ introduced in the section Results, it is useful to introduce the coefficient $D_R^{(in)} = \sum f_n(0)b_n(L)/Q$. Both $|D_R|$ and $|D_R^{(in)}|$ lies in the range 0 to 1. $D_R(L)$ depends on the output amplitudes $f_n(L)$ and is therefore unknown. On the other hand, $D_R^{(in)}$ represents the correlation among the input forward and input backward mode amplitudes and is therefore fixed by the boundary conditions. The cornerstone of our theoretical analysis consists in the derivation of a relation between $D_R(L)$ and $D_R^{(in)}$. When the Kerr coefficients $\gamma_{nn} = \gamma$ are the same and $\gamma_{nm} = (1/2)\gamma$, Eq. (1) turn out to be integrable. Under this condition, starting from Eq. (1) the following relation is found (details of the mathematics are provided in the Supplementary information note 1):

$$\sin^2(Th) = \frac{\left(h^2 - |D_R^{(in)}|^2 - v^2\right) \cdot h^2}{\left(h^2 - v^2\right)\left(h^2 - w^2\right)} \qquad (3)$$

Where $v = \Delta P/(2Q)$; $w = P_{tot}/(2Q)$; $h^2 = v^2 + |D_R(L)|^2$, being $P_{tot} = P_f + P_b$ the total power, $\Delta P = P_b - P_f$ the differential backward-forward power; $T = (T_f \, T_b)^{1/2}$. The quantity $T_f = L\gamma P_f$ indicates the total number of nonlinear lengths for the FS. Similarly, $T_b = L\gamma P_b$ represents the total number of nonlinear lengths for the BCB. The geometric average $T = (T_f \, T_b)^{1/2}$ is therefore a key parameter, as it can be interpreted as an indicator of the overall system nonlinearity.

Equation (3) can be solved graphically, which provides a clear understanding of the most suitable configurations under which effective mode rejection can be achieved. When the system nonlinearity is substantial and the forward and backward power are equally distributed ($\Delta P \sim 0$), we find that $|D_R(L)| \sim \mathrm{asin}(|D_R^{(in)}|)/T$ (see Supplementary information note 1) and then ultimately $D_R(L) \sim 0$ in a strongly nonlinear regime ($T \gg 1$). As anticipated in the section Results, this implies effective rejection of mode-$m$ when the BCB is coupled to that mode.

### Theory of mode attraction

Starting from Eq. (2) we can elaborate a theory that explains the mode attraction process, which follows similar steps to those previously illustrated in the case of mode rejection. We first introduce the correlation coefficients $D_A(L) = \sum f_n(L)b_n(L)^*/Q$ and $D_A^{(in)} = \sum f_n(0)b_n(L)^*/Q$, which are the counterpart of the coefficients $D_R(L)$ and $D_R^{(in)}$ used in the framework of mode rejection. Finally, from Eq. (2) we derive the following relation between $D_A(L)$ and $D_A^{(in)}$ (details of the mathematics in the Supplementary information note 1):

$$\sin^2(Th) = \frac{\left(h'^2 + |D_A^{(in)}|^2 - w^2\right) \cdot h'^2}{\left(h'^2 - v^2\right)\left(h'^2 - w^2\right)} \qquad (4)$$

Where $h'^2 = w^2 - |D_A(L)|^2$. Equation (4) can be solved graphically. When the system nonlinearity is substantial and the forward and backward power are equally distributed ($\Delta P \sim 0$), then $|D_A(L)| \sim [1 - \mathrm{acos}(|D_A^{(in)}|)^2/T^2]^{1/2}$, therefore $|D_A| \sim 1$ in a strong nonlinear regime ($T \gg 1$). As anticipated in the section Results, this implies effective attraction of the output FS towards the mode of the input BCB.

### Multicore fibre fabrication

The DCF and TCF used in this work have a core size of ~5 μm, a numerical aperture of ~0.15 and a core-to-core distance of ~9.5 μm (see Supplementary information note 3). The core disposition in both the DCF and TCF confers a substantial birefringence (>+10 dB polarisation extinction ratio), which allow for effective polarisation maintenance. Both these fibres were fabricated via a stack and draw

process in the cleanrooms of the University of Southampton. Initially a series of doped and pure fused silica rods were drawn to precise dimensions matching a carefully designed stack plan. The pure fused silica rods were drawn from Heraeus F300 glass whereas the doped glass rods came from a commercial preform fabricated externally (Prysmian) to an in-house generated design. All the rods used were drawn on our fibre drawing tower with extreme care taken to ensure their outside diameter (OD) was precisely controlled and its variation minimised. The rods were then stack inside an F300 tube such that the interstitial gaps between them were minimised to reduce the likelihood of any structural deformation during the fibre drawing process when these gaps are removed via the application of a vacuum. The DCF and TCF were drawn from different stacks, due to the differing fibre geometries. To ensure the cleanliness of both stacks they were built in a class 100 cleanroom area, with care taken in the handling of the rods to minimise the possibility of contamination, and then carefully prepared on a glass working lathe prior to fibre drawing. This was done to ensure the final fibre is free from any internal or external contaminants (that could disrupt the fibres' structure or optical properties) and reduce the likelihood of breakage during fibre drawing, characterisation, and use. The fibres were coated with a standard UV-cured high refractive index polymer coating (Desolite DSM-314) both to protect the fibre and to help strip any light that may inadvertently be launched into the fibres jacket glass during its use.

### Experiments

We conducted experiments by launching ~500 ps long pulses at a wavelength of 1040 nm from an in-house all-fiberized ytterbium master oscillator power amplifier with linearly-polarised output at the two ends of the fibres under test. The coupled power and polarisation of the FS and the BCB are tuned with a suitable combination of polarisation beam splitters and wave plates. A water-cooling spatial light modulator (Holoeye PLUTO-2-NIR-149) is employed to control the coupling of the BCB, whereas in the case of the FS a spatial phase-plate is used to excite an arbitrary combination of modes. The coupling conditions are controlled by optimising the phase pattern on the SLM and by adjusting the position of phase plate using a precision three-axis stage.

The output from each end of the fibres under test is sampled by a wedge and the near- and far-field beam profiles are imaged on a CCD camera. The near-and far-field profiles are then used to estimate the mode content of the FS and BCB via mode decomposition[50]. By numerically calculating the mode weight and relative phase in an iterative process (Stochastic Parallel Gradient Descent algorithm is successfully applied[51]), a reconstructed spatial distribution is obtained and is compared with the measured spatial profile to iteratively optimise the mode decomposition results. The reconstructed spatial distribution has a correlation factor of ~99% with respect to the measured profile for different mode combinations and different fibres (see Supplementary information note 4 and related Supplementary movies 1–9 for details).

### Data availability
The data are available at ref. 52.

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

## Acknowledgements

M.G. acknowledges funding from the European Research Council under the H2020 Programme (ERC Starting Grant No. 802682, MODES project) and from the UK Engineering and Physical Sciences Research Council (EP/T019441/1); K.J. acknowledges funding from China Scholarship Council (202006840003); J.S. and D.J.R. acknowledges funding from the UK Engineering and Physical Sciences Research Council (EP/P030181/1); S.W. acknowledges funding from the European Research Council under the H2020 Programme (ERC Advanced Grant No.740355, STEM project) and from European Union- Next Generation EU (PE00000001, RESTART).

## Author contributions

K.J. performed all the experiments reported in this work and performed the numerical simulations with M.G.; I.D. and J.S. fabricated the multi-core fibres employed in the experiments; D.J.R. provided support for the experimental work and together with M.G. supervised the project; S.W. provided support for the theoretical work and contributed to the interpretation of the results; M.G. additionally conceived the research idea, developed the theoretical results and wrote the paper with feedback from all the authors.

## Competing interests

The authors declare no competing interests.
