## [Peer Review File · Nature Communications]

Mode attraction, rejection and control in nonlinear multimode opticsREVIEWER COMMENTS

Reviewer #1 (Remarks to the Author):

This manuscript by Ji and co-workers reports on a theoretical, numerical and experimental study of nonlinear spatial mode control (mostly a rejection effect) occurring in multicore/multimode fibers between an arbitrary spatially distributed forward signal (FS) and a specific (mainly a unique mode) counter-propagating beam (BCB).

The concept described here is not completely novel. Indeed, the main idea of this paper is a generalization of the concept of spatial mode control with $N > 2$ introduced and experimentally demonstrated in a bimodal fibre in ref. [34]. See for instance Fig. 2 of ref. [34] compared to present Fig. 3a. However, the authors provide here a great piece of work by demonstrating the general case of $N > 2$, whilst highlighting the effect of mode rejection, as well as preliminary results of mode control (section sup-V).

The experimental, numerical and theoretical analysis are quite convincing. These results represent a step forward regarding the control of light beams in multimode fibers and would deserve publication in a good journal. However, the paper is sometimes hard to follow and the main message i.e., mode rejection and control is often blurred through out the manuscript. I would suggest the authors to strongly revise their manuscript and focus only on the mode control (the difference between attraction and rejection could be provided in SI, whilst sup-V should be included in the main article).

More precisely, the justifications of this research with very general sentences as well as long sections dedicated to the difference between attraction and rejection blur the main results. Especially, the comparison of the 2 models described respectively by Eqs. (1) i.e., the "real" model related to the rejection experiment demonstrated here and Eqs. (2), which describe theoretically the attraction phenomenon in-between N circular components of polarization – for which no real system seems to exist – hinders the reader to catch the true novelty and main results.

More technical points:

1. The authors should introduce earlier the fact that they mostly use PM multicore fibres and associated super-modes, which represent quite specific types of optical fibres.
2. The authors should discuss and show the BCB spatial profile together with the FS profile. Indeed, since both waves interact and reach a stationary regime, it becomes important to monitor both counter-propagating beam profiles.
3. In Fig. 1d, could the authors explain why does the output FS contain 50% in M2 and 50% in M3?
4. In Fig. 2b, there is a typo for the profile of LP21b.
5. Why the input power of the FS is different in Fig. 3(a-c)?
6. For applications, it is important to consider longer fibers? What does happen if this phenomenon occurs in a 10-m or even 100-m long fibre?
7. An important issue underlined by the authors is the polarization states of the two counter-propagating beams. Could the authors comment this point and show some results in non-PM standard multimode fibers?
8. The authors used the same laser for generating both counter-propagating beams. Could the authors comment the practical possibility of using two different sources? Is the phenomenon still working if a difference of wavelength is introduced between the FS and BCB beams?
9. The authors used home-made multi-core fibres. Is linear mode coupling (in-between adjacent cores) should be taken into account?
10. In Fig. 4, the fibre perturbations remain barely explained, could the authors provide more details? Is it corresponding to different input conditions?
11. Through out the figures, the authors should add some error-bars for the relative power contains in the different modes of the output FS.

Reviewer #2 (Remarks to the Author):

The authors consider effects related to counter-propagating waves in multimode waveguides, that are indeed, are not explored yet. An interesting new phenomenon was observed that is coined rejection of the modes in multimode fiber. The consideration of the rejection model is done using the analogy with the model of mode attraction in anisotropic and birefringent fibers. I recommend the paper for publication, but suggest to address the following points in the revision:

1) It is stated that if Q and $z = L$ are large enough. The logic of constructing the correlation coefficient $DR(z)$ and explaining the mode rejection could be expanded to make it more clear to general readers. In the main manuscript it is written:

"In the case of Eqs.(1), by calling P_f (P_b) the total injected FS (BCB) power and $DR(z)$ the normalised correlation coefficient $DR(z) = \sum f_n(z)b_n(z)/Q$ with $Q=(P_f P_b)^{1/2}$, one finds that $DR(L)$ decreases as $1/(L \cdot Q)$: therefore, $|DR(L)| \rightarrow 0$ for sufficiently large L or Q . Mode rejection is a direct consequence of this property."

I found this explanation a bit confusing. Isn't it by definition given above $DR(z) \rightarrow 0$ at sufficiently large Q ?

"...Indeed, if the input BCB is launched into an individual mode only, say the m -mode, such that $|b_m(L)| = P_b^{1/2}$ and $b_n(L) = 0 \forall n \neq m$, then $|DR(L)|$ reduces to $|f_m(L)|/P_f^{1/2}$. The condition $|DR(L)| \rightarrow 0$ implies therefore that $f_m(L) \sim 0$: namely, irrespective of its input mode distribution, the output FS carries no energy in the m -mode, that is to say the m -mode is rejected at the output..."

Is L a launch point?

In the Supplementary:

"In the following, our focus is on the derivation of an equation for the correlation coefficient $DR = (\sum_n f_n b_n) / Q$, with $Q = (P_f P_b)^{1/2}$..."

Good, then just find how $D_R(z)$ evolves and only after that put the comments.

I would suggest to make it much more clear without a chance for alternative interpretations.

2) Does the transverse distributions of the field of the modes impact the conclusion about rejection?

3) Several fibers are shown in Fig. 2b. Is it possible to observe the similar effect experimentally in the ordinary multimode fibers? If the fields are \sim where n is the number of zeros along the radial direction, will this degree of freedom affect the general conclusion about correlation coefficient?

4) The correlation coefficient DR can include several terms of the sum, then the logic of rejection these several modes f_m in the sum will be the same as for one mode?

All should be better explained.

Reviewer #3 (Remarks to the Author):

In this paper the authors consider nonlinear interactions in multimode optical fibers (MMF). In particular the main result is in the case of seeding a particular mode in a counterpropagating format, it universally suppresses or inhibits energy being propagated in on this mode in the forward propagating light. This gives the term "mode rejection". In an alternative fiber configuration they instead found "mode attraction". While it is the case that there is much recent work on MMF in terms of complex dynamics under the general term "of an initial value problem" (seeding in one end) and going years back, there were studies of using counterpropagating schemes for effective energy transfer (more notably I would say in quadratic media) which is a more difficult "boundary value problem". I agree with the authors that this work presents a somewhat new paradigm with potential applications.

The results rely on asserting the validity of the model (eqs. 1, 2) which is done with experiments on different fibers, so while it may be interesting to look at this problem for the full Maxwell's eq (no mode decomposition), it seems the results are robust.

I don't have much to add other than in my opinion this configuration and for the general nonuniform values of the coupling parameters is prime to be studied under a Physics-informed neural networks (PINN) setting. I'll let the authors comment on this suggestion.

We are glad that all the 3 reviewers have provided a positive assessment on the relevance of the ideas as well as on the quality of the results reported in our manuscript. We are sincerely grateful to them for the time they spent in their review. They have provided important and useful suggestions to reshape the manuscript. We have carefully considered all their comments and we believe this has improved the overall quality of the manuscript. Here below we report a point-by-point reply to the comments. We have highlighted in bold the sentences that should help the reviewers to locate the changes in the revised manuscript. In the revised manuscript, these changes are highlighted in red.

REVIEWER COMMENTS

Reviewer #1 (Remarks to the Author):

This manuscript by Ji and co-workers reports on a theoretical, numerical and experimental study of nonlinear spatial mode control (mostly a rejection effect) occurring in multicore/multimode fibers between an arbitrary spatially distributed forward signal (FS) and a specific (mainly a unique mode) counter-propagating beam (BCB).

The concept described here is not completely novel. Indeed, the main idea of this paper is a generalization of the concept of spatial mode control with $N > 2$ introduced and experimentally demonstrated in a bimodal fibre in ref. [34]. See for instance Fig. 2 of ref. [34] compared to present Fig. 3a. However, the authors provide here a great piece of work by demonstrating the general case of $N > 2$, whilst highlighting the effect of mode rejection, as well as preliminary results of mode control (section sup-V). The experimental, numerical and theoretical analysis are quite convincing. These results represent a step forward regarding the control of light beams in multimode fibers and would deserve publication in a good journal.

However, the paper is sometimes hard to follow and the main message i.e., mode rejection and control is often blurred through out the manuscript. I would suggest the authors to strongly revise their manuscript and focus only on the mode control (the difference between attraction and rejection could be provided in SI, whilst sup-V should be included in the main article).

More precisely, the justifications of this research with very general sentences as well as long sections dedicated to the difference between attraction and rejection blur the main results. Especially, the comparison of the 2 models described respectively by Eqs. (1) i.e., the “real” model related to the rejection experiment demonstrated here and Eqs. (2), which describe theoretically the attraction phenomenon in-between N circular components of polarization – for which no real system seems to exist – hinders the reader to catch the true novelty and main results.

We agree that the information provided in Sup-V should be included in the main manuscript, as it provides an important overview on the applications of the theoretical model developed in our manuscript. **As suggested by the reviewer, the information previously reported in Sup-V is now described in the section Results (subsection “Towards the ideal of mode control”) of the main manuscript.**

Because the notion of *attraction* is well known and central in physics, we believe that discussing the contrast and differences between attraction and rejection helps the reader to appreciate the relevance of both. Therefore, we believe the comparison attraction-rejection remains a central message to deliver in the main manuscript. Note also that Eqs.(2) describes a real model for $N=2$ (polarisation attraction in single-mode fibres). At present, fibre systems where Eqs.(2) would apply with $N > 2$ are unknown, but they may still exist. This is all the more true if one extends the research to other physical systems beyond optical fibres. **We have added a paragraph in the section Discussion to further emphasize this point:**

“The scope of this generalisation is two-fold. Firstly, it allows a direct comparison between Eqs.(1) and (2), and therefore between the fundamental notions of rejection and attraction, in systems of arbitrary dimension N . Second, it sheds light on the universality of these processes, which goes beyond bidimensional systems. Although currently it is yet unknown the existence of multimode systems where Eqs.(2) with $N > 2$ applies, however these may be found in the future, either in optical fibres or different physical systems.”

More technical points:

1. The authors should introduce earlier the fact that they mostly use PM multicore fibres and associated super-modes, which represent quite specific types of optical fibres.

We note that the use of multicore fibres is mentioned as early as in the abstract (last sentence) and in the Introduction (line 81). **As suggested by the reviewer we have explicitly mentioned in the abstract that these are polarization-maintaining fibres.**

It is important however to not overemphasize the use of multicore fibres. Indeed, our outcomes are independent of the type of multimode platform, provided that polarization is preserved. In this work we have demonstrated mode rejection in 2 multicore fibres (Fig.2b, left and central fibres) as well as in 1 standard commercial multimode/single-core fibre (Thorlabs fibre on the right side of Fig.2b).

2. The authors should discuss and show the BCB spatial profile together with the FS profile. Indeed, since both waves interact and reach a stationary regime, it becomes important to monitor both counter-propagating beam profiles.

We thank the reviewer for pointing this out. We agree that the BCB dynamic represents important information that was missing in the original manuscript. **In order to fully address this issue, we have added a new subsection named “Dynamic of the backward control beam” in the section Results, which includes a description of the BCB dynamic as well as some new experimental result where the BCB is analysed.** It is worth noting that these experimental results fit well our simulations, which provides further evidence of the validity of our theoretical and numerical results.

3. In Fig. 1d, could the authors explain why does the output FS contain 50% in M2 and 50% in M3?

Fig.1d has illustrative purpose only, namely, showing the rejection of mode M1 (so the output FS has 0% in M1). In this case the values of M2 (50%) and M3 (50%) are purely fictitious. In a real experiment, while mode M1 would be consistently rejected (that is, output FS contains ~ 0 % in M1) irrespectively of the modal state of the input FS, however the specific output value for M2 and M3 would depend on the modal state of the input FS and the fibre parameters. Basing on this reviewer’s observation, we realised that this figure could be misleading as it may induce the reader to believe that equipartition of energy is achieved in M2 and M3. **We have therefore modified accordingly Fig.1 and the related caption.**

4. In Fig. 2b, there is a typo for the profile of LP21b.

We thank the reviewer. **This has been now corrected.**

5. Why the input power of the FS is different in Fig. 3(a-c)?

The experiments reported in the 3 panels of Fig.3 were carried out at different times. Since we are using a free space setup, careful realignment is required each time, which leads to slight variations in the coupling efficiency. This explains the small differences in the input FS power, such as those between Fig.3b (6.5 kW) and Fig.3c (6.2 kW). On the other hand, in the case of Fig.3a the power is deliberately lower because we wanted to demonstrate mode rejection in a weaker nonlinear regime. **We have added the following paragraph in the Supplementary Sup-III (and modified Table S5 accordingly) to clarify the choice of the power levels employed in our experiments:**

“The values of peak power have been chosen to achieve a high nonlinear regime, which is necessary condition to observe the mode rejection dynamics. As reported in the section “Methods-Theory of mode rejection”, the parameter $T=(T_f T_b)^{1/2} = L\gamma(P_f P_b)^{1/2}$ represents the overall system nonlinearity. As a rule of thumb, our simulations indicate that we need $T>2$ to observe effective mode rejection. In the fibres under test, where the Kerr coefficients are generally different each other, γ is replaced by their arithmetic average γ_{av} . Moreover, because we use pulsed beams, the fibre length L is replaced by the actual interaction length $L_{in}=2\cdot t_0\cdot c$ (c =speed of light in the fibre, t_0 =pulse width, 0.5 ns). In conclusion, the parameter $T_{eff} = L_{in}\gamma_{av}(P_f P_b)^{1/2}$, reported in the last row of Table S5, is used to evaluate the system nonlinearity. The peak powers P_f and P_b are properly chosen so as to meet the condition $T_{eff}>2$.”

We have also modified the sentence in the main text, line 173-174, as follows:

“Details on the fibres and experimental parameters are provided in Fig.2b and Supplementary section Sup-III, including the choice of the optical power to achieve the high levels of system nonlinearity required.”

6. For applications, it is important to consider longer fibers? What does happen if this phenomenon occurs in a 10-m or even 100-m long fibre?

As mentioned in our previous reply, the key-parameter is the overall system nonlinearity, which is proportional to the actual interaction length L_{in} .

In the case of pulsed beams $L_{in}=2\cdot t_0\cdot c$, therefore the system nonlinearity depends on the pulse width t_0 rather than the fibre length L . Indeed, we performed experiments with fixed pulse width $t_0 = 0.5$ ns and fibre lengths in the range 0.4-4 m without observing any meaningful difference.

In the case of continuous-wave (CW) beams the counter-propagating fields interact in each point of the fibre, that is to say, $L_{in}=L$. In other words, with CW beams the total system nonlinearity is proportional to the fibre length L . Therefore, for the same power levels a longer fibre leads to a stronger mode rejection effect.

Alternatively, one may relax the power requirements, e.g. reducing the power of a factor 10 and then increasing the fibre length of the same factor without compromising the mode rejection.

We stress that the above-mentioned considerations are strictly valid in the absence of additional parasitic effects, e.g. Raman scattering, losses or group-velocity dispersion, that may be exacerbated in longer fibres. As explained in the Methods section, these effects are negligible in the short fibres used in our experiments.

7. An important issue underlined by the authors is the polarization states of the two counter-propagating beams. Could the authors comment this point and show some results in non-PM standard multimode fibers?

Our theoretical model is valid whenever the counter-propagating beams are either co-polarized or orthogonally polarised. We have recently obtained some results in non-PM fibres (see Ref.[39]), however these preliminary results do not meet the aforementioned criteria on the polarization conditions. Indeed, we have later realised that in non-PM fibres polarization attraction interplays with mode rejection. This not only prevents from scaling up our analysis to more than 2 modes, but also prevents from maintaining the relative polarization between FS and BCB, which may finally lead to some ambiguity in our measurements. For this reason in this work we moved to PM fibres. By maintaining the polarization, we not only were able to build the theoretical foundations of the rejection principle in N-dimensional systems (N arbitrary), but could also perform experiments where mode rejection is observed unambiguously in multimode fibres supporting up to 6 modes.

We have mentioned Ref.[39] in the Introduction (line 65). Note that at the time when Ref.[39] was published we were not aware of the rejection principle and therefore we named erroneously the process as mode attraction (as in Ref.[34]).

8. The authors used the same laser for generating both counter-propagating beams. Could the authors comment the practical possibility of using two different sources? Is the phenomenon still working if a difference of wavelength is introduced between the FS and BCB beams?

The reviewer raised an interesting point that deserves to be properly discussed in the manuscript. Using different sources for the FS and the BCB is certainly possible. We could not test this case as we have only one high-power source available, however some basic conclusion can be drawn from our theoretical model. **To clarify this point, we have added the following paragraph in the Section “Methods- Theoretical framework and numerical simulations”:**

“We note that if FS and BCB are centred at different carrier wavelengths, say λ_F and λ_B respectively, then the intermodal power exchange interactions (last terms on the right-hand-side of Eqs.(1,2)) are subject to a nonlinear phase mismatch $\Delta\beta_{nm} = \beta_n(\lambda_F) - \beta_n(\lambda_B) + \beta_m(\lambda_B) - \beta_m(\lambda_F)$, where $\beta_n(\lambda)$ is the propagation constant of modes n at wavelength λ . The overall dynamics discussed in our manuscript remain unchanged whenever the interaction length L_{in} is much shorter than the corresponding beat length $2\pi/\Delta\beta_{nm}$. By expanding in Taylor series the phase-mismatch $\Delta\beta_{nm}$, we find after some algebra that the condition $L_{in} \ll 2\pi/\Delta\beta_{nm}$ can be recast as follows: $\Delta\lambda \ll \lambda_0^2 / (c L_{in} \Delta\beta_{1,nn})$, where $\Delta\lambda = \lambda_F - \lambda_B$, $\lambda_0 = (\lambda_F + \lambda_B)/2$ is the central wavelength and $\Delta\beta_{1,nn}$ is the differential inverse group velocity between modes n and m . In the case of the fibres under test, whose parameters are reported in Supplementary Sup-III, we find $\Delta\lambda \ll 25$ nm, therefore the system dynamics would be unaffected if FS and BCB are detuned a few nanometres apart.”

9. The authors used home-made multi-core fibres. Is linear mode coupling (in-between adjacent cores) should be taken into account?

We stress that our theoretical model and simulations are not based on the individual core fields, but on the actual fibre modes (called supermodes in the case of multi-core fibres). This is indeed the most accurate way to describe the linear dynamics. The situation would be different if, instead of the supermodes, we used the

individual core fields. However, in that case Eqs.(1,2) would be substantially different, and would require to include the linear coupling between the different cores, as mentioned by the reviewer.

10. In Fig. 4, the fibre perturbations remain barely explained, could the authors provide more details? Is it corresponding to different input conditions?

We have added the following sentence in the caption of Fig.4 to clarify this issue. “The fibre is perturbed 5 different times. Each time, the perturbation consists in compressing or bending the fibre with different levels of intensity and in different points.”

11. Through out the figures, the authors should add some error-bars for the relative power contains in the different modes of the output FS.

We agree with the reviewer on this point. **We have redone all the figures with the error bars.** Because adding the bars to each mode would create confusion in some of the figures, we have added the bars only to the mode that is rejected, which is the focus of our experimental investigation. **We have also added the following sentence in the section Results (subsection “Experimental results on mode rejection”): “In Fig.3 and in the next figures, error bars of $\pm 3\%$ were added to the measured relative power of the rejected mode, which represents the estimated uncertainty of our mode decomposition algorithm”.**

Reviewer #2 (Remarks to the Author):

The authors consider effects related to counter-propagating waves in multimode waveguides, that are indeed, are not explored yet. An interesting new phenomenon was observed that is coined rejection of the modes in multimode fiber. The consideration of the rejection model is done using the analogy with the model of mode attraction in anisotropic and birefringent fibers. I recommend the paper for publication, but suggest to address the following points in the revision:

1) It is stated that if Q and $z = L$ are large enough. The logic of constructing the correlation coefficient $DR(z)$ and explaining the mode rejection could be expanded to make it more clear to general readers. In the main manuscript it is written:

“In the case of Eqs.(1), by calling P_f (P_b) the total injected FS (BCB) power and $DR(z)$ the normalised correlation coefficient $DR(z) = \sum f_n(z)b_n(z)/Q$ with $Q=(P_f P_b)^{1/2}$, one finds that $DR(L)$ decreases as $1/(L \cdot Q)$: therefore, $|DR(L)| \rightarrow 0$ for sufficiently large L or Q . Mode rejection is a direct consequence of this property.”

I found this explanation a bit confusing. Isn't it by definition given above $DR(z) \rightarrow 0$ at sufficiently large Q ? “...Indeed, if the input BCB is launched into an individual mode only, say the m -mode, such that $|b_m(L)| = P_b^{1/2}$ and $b_n(L) = 0 \forall n \neq m$, then $|DR(L)|$ reduces to $|f_m(L)|/P_f^{1/2}$. The condition $|DR(L)| \rightarrow 0$ implies therefore that $f_m(L) \sim 0$: namely, irrespective of its input mode distribution, the output FS carries no energy in the m -mode, that is to say the m -mode is rejected at the output....”

Is L a launch point?

In the Supplementary:

“In the following, our focus is on the derivation of an equation for the correlation coefficient $DR = (\sum_n f_n b_n) / Q$, with $Q = (P_f P_b)^{1/2}$...”

Good, then just find how $D_R(z)$ evolves and only after that put the comments.

I would suggest to make it much more clear without a chance for alternative interpretations.

We understand the concern of the reviewer about the definition of $D_R(z)$. First of all, let us indicate with $D_{R_NUM} = \sum f_n(z)b_n(z)$ the numerator of $D_R(z)$. We note that D_{R_NUM} is a function of P_f and P_b , so let us indicate it as $D_{R_NUM}(P_f, P_b)$. This is because the conditions $P_f = \sum_n |f_n|^2$ and $P_b = \sum_n |b_n|^2$ hold true. Therefore, we can write $D_R = D_{R_NUM}(P_f, P_b) / (P_f P_b)^{1/2}$.

It should be clear now that the condition $P_f \rightarrow \infty$ and/or $P_b \rightarrow \infty$ does not necessarily imply $D_R \rightarrow 0$. In other words, the value of D_R for large P_f and P_b strictly depends on how $D_{R_NUM}(P_f, P_b)$ evolves as a function of P_f and P_b . **In order to clarify this point, we have explicitly reported the above-mentioned conditions $P_f = \sum_n |f_n|^2$ and $P_b = \sum_n |b_n|^2$ in the main manuscript, see line 130.**

As for the position L , it is actually the launching point for the BCB and the exit point for FS, see e.g. Fig.2a and the main text line 113: “...The FS (BCB) enters the fibre at $z=0$ ($z=L$)...”.

In order to further clarify this point we have modified the sentence mentioned by the reviewer as follows, see line 135-136 “...namely, irrespective of the mode distribution of the input FS in $z=0$, the output FS in $z=L$ carries no energy in the m -mode...”

2) Does the transverse distributions of the field of the modes impact the conclusion about rejection?

No, because our theory is independent of the specific transverse shape of the modes. In our experiments reported in Figs.3,5 and 6 we have proved rejection for different kind of modes having different spatial shapes.

3) Several fibers are shown in Fig. 2b. Is it possible to observe the similar effect experimentally in the ordinary multimode fibers? If the fields are \sim where n is the number of zeros along the radial direction, will this degree of freedom affect the general conclusion about correlation coefficient?

We have observed mode rejection in the Thorlabs fibre PM-6MF displayed in Fig.2b, which is indeed an ordinary commercial polarization maintaining fibre. As pointed out in the Section Methods, we believe that mode rejection would be observed in any step-index polarization maintaining fibre, and more in general any structure with step-index birefringent geometry, e.g. silicon on insulator waveguides. On the other hand, Eqs.(1) does not describe other kind of ordinary fibres, like for example multimode GRIN fibres. In these fibres polarization is not maintained and the refractive index profile is not step-index. Consequently, they are affected by substantial polarization and mode coupling. It is unclear whether and what kind of mode rejection dynamics would be observed in this case. This is currently under investigation.

As for the correlation coefficients, our derivation reported in the Supplementary Sup-I is independent of the spatial shape of the modes into play (including the number of zeros in the radial direction), therefore the general conclusion on the correlation coefficient is unchanged.

4) The correlation coefficient DR can include several terms of the sum, then the logic of rejection these several modes f_m in the sum will be the same as for one mode? All should be better explained.

Here the reviewer raises an important point. In general, if the BCB is coupled to one mode only, then the output FS undergoes rejection independently of the modal state of the input FS. On the other hand, if the BCB is coupled to several modes, new scenarios and related applications open up. We had introduced in the Supplementary Sup-V some information on this regard. Given this reviewer's comment (and, similarly, first comment of reviewer #1), we realised this issue is important and should be included in the main manuscript.

Therefore, the information previously reported in Sup-V is now described in the Section Results (subsection “Towards the ideal of mode control”) of the revised manuscript. Hopefully this should help the reader understanding the dynamic when the BCB is coupled to a combination of modes. It should be noted however that this is not an exhaustive analysis, yet, but rather preliminary (see Fig.8) and therefore further investigation will be needed to fully understand the practical applications and limits.

Reviewer #3 (Remarks to the Author):

In this paper the authors consider nonlinear interactions in multimode optical fibers (MMF). In particular the main results is in the case of seeding a particular mode in a counterpropagating format, it universally suppresses or inhibits energy being propagated in on this mode in the forward propagating light. This gives the term "mode rejection". In an alternative fiber configuration they instead found "mode attraction". While it is the case that there is much recent work on MMF in terms of complex dynamics under the general term "of an initial value problem" (seeding in one end) and going years back, there were studies of using counterpropagating schemes for effective energy transfer (more notably I would say in quadratic media) which is a more difficult "boundary value problem". I agree with the authors that this work presents a somewhat new paradigm with potential applications.

The results rely on asserting the validity of the model (eqs. 1, 2) which is done with experiments on different fibers, so while it may be interesting to look at this problem for the full Maxwell's eq (no mode decomposition), it seems the results are robust.

I don't have much to add other than in my opinion this configuration and for the general nonuniform values of the coupling parameters is prime to be studied under a Physics-informed neural networks (PINN) setting. I'll let the authors comment on this suggestion.

We thank the reviewer for the positive assessment of this manuscript and for the suggestions.

We definitely agree that it would be useful to simulate the problem via full Maxwell equations and starting from an input FS with a speckled transverse profile (smoothly randomized in space). This may provide further insights into the dynamic and we believe it would also provide further confirmation of the mode rejection/attraction principles introduced in this work. A 3D simulation is out of reach for the computational resources we have at disposition, however we are currently attempting a 2D simulation by making some preliminary assumption (e.g. input FS having circular symmetry).

Unfortunately, we have no background knowledge on PINNs, therefore we cannot provide a specific opinion. Actually, we have discovered the PINNs thanks to this comment and we understand it may have a great potential to demonstrate theoretically the mode rejection/attraction principles in the general case of non-uniform Kerr coefficients. We are therefore sincerely grateful for the reviewer's suggestion. Luckily enough, we have colleagues that are expert in neural networks and that may provide help on this.

REVIEWER COMMENTS

Reviewer #1 (Remarks to the Author):

The authors have strongly improved their manuscript, whilst addressing all my previous concerns. The new version is clearer and the overall text flow looks smoother. In particular, the comparison between attraction and rejection effects as well as the generalization to N modes now appear more natural and as a central message.

The addition of the new results regarding the evolution of the back-propagating beam and depicted in the novel Fig. 7 reinforces the whole scenario. These complementary measurements clearly underline that this system of 2 counter-propagating beams converges towards a unique and mutual steady-state arrangement.

Furthermore, the inclusion of the section dedicated to the generalization of these effects towards a possible mode control (former Sup-V) now fully unveils the potential of this peculiar nonlinear dynamics for future applications.

In terms of applications, the authors have also greatly discussed the practical case for which different laser sources are used to generate the forward and backward signals.

For all the reasons described above, I fully support the publication of this revised version of the manuscript providing the authors address the following minor point.

My final question deals with the use of sub-ns pulses (shorter than the fibre length and resulting in tens of cm of interacting length), instead of CW. Did the authors observe some transient regimes (by recording the output profile or intensity as a function of time, as briefly mentioned in Sup-2)? Asymmetries between the leading and tailing fronts? Indeed, it is known in the counter-propagating polarization attraction process that the system needs a certain time to reach a stationary regime, which mainly depends on the amount of nonlinearity (here described as $T > 2$ probably). Some examples of transient regime have already been observed by the group of Prof. Wabnitz in the following paper [Kozlov, et al., JOSA B 28, 1782 (2011)]. Since the authors use sub-ns pulses, instead of continuous waves, and that the system is characterized by a well-defined response-time, it could be helpful for the reader to mention that point. Especially, the authors could mention the behaviour of the system when the incoming beam is perturbed or fluctuates in input at a rate close to the response-time of the system.

Reviewer #2 (Remarks to the Author):

I am happy with the revision made by the authors and recommend to publish paper as it is now.

REVIEWER COMMENTS

Reviewer #1 (Remarks to the Author):

The authors have strongly improved their manuscript, whilst addressing all my previous concerns. The new version is clearer and the overall text flow looks smoother. In particular, the comparison between attraction and rejection effects as well as the generalization to N modes now appear more natural and as a central message.

The addition of the new results regarding the evolution of the back-propagating beam and depicted in the novel Fig. 7 reinforces the whole scenario. These complementary measurements clearly underline that this system of 2 counter-propagating beams converges towards a unique and mutual steady-state arrangement.

Furthermore, the inclusion of the section dedicated to the generalization of these effects towards a possible mode control (former Sup-V) now fully unveils the potential of this peculiar nonlinear dynamics for future applications.

In terms of applications, the authors have also greatly discussed the practical case for which different laser sources are used to generate the forward and backward signals. For all the reasons described above, I fully support the publication of this revised version of the manuscript providing the authors address the following minor point. My final question deals with the use of sub-ns pulses (shorter than the fibre length and resulting in tens of cm of interacting length), instead of CW. Did the authors observe some transient regimes (by recording the output profile or intensity as a function of time, as briefly mentioned in Sup-2)? Asymmetries between the leading and tailing fronts? Indeed, it is known in the counter-propagating polarization attraction process that the system needs a certain time to reach a stationary regime, which mainly depends on the amount of nonlinearity (here described as $T > 2$ probably). Some examples of transient regime have already been observed by the group of Prof. Wabnitz in the following paper [Kozlov, et al., JOSA B 28, 1782 (2011)]. Since the authors use sub-ns pulses, instead of continuous waves, and that the system is characterized by a well-defined response-time, it could be helpful for the reader to mention that point. Especially, the authors could mention the behaviour of the system when the incoming beam is perturbed or fluctuates in input at a rate close to the response-time of the system.

We are sincerely grateful to the reviewer for his appreciation on the new version of the manuscript. This comes as the result of the useful comments from the reviewers, which has allowed us to improve the overall quality of this work.

It is correct that the system experiences a transient regime before achieving the stationary regime, as reported in the Supplementary figure S2(b). On the other hand, as it can be seen in that figure, given the level of fibre nonlinearity and the level of power employed in the experiments to achieve relevant mode rejection, the transient time T_{TRANS} is of the order of tens of nanoseconds. Unfortunately, this prevents us to observe the transient regime with the experimental setup that we have at disposition. Indeed, our mode decomposition technique is based on the acquisition of images using the infrared camera, whose integration time is several orders of magnitudes larger than T_{TRANS} .

With regard to fast input perturbations, it should be noted that differently from single-mode fibres, where the system response time is defined as $c/(\gamma P_f)$ (c = speed of light into the fibre, γ = Kerr coefficient, P_f = forward power), in the case of multimode fibres the definition is ambiguous, as there are several and different intramodal and intermodal Kerr coefficients. A reasonable estimate is assuming the response time T_{RES} of the multimode system being $T_{\text{RES}} = c/(\gamma_{\text{avg}} P_f)$, where γ_{avg} is the average Kerr coefficient. In the fibres under test T_{RES} is sub-nanosecond. Once again, this prevents any experimental observation with the current setup.

We have run numerical simulations that seems to indicate that fast fluctuations (burst) of the input signal over a time scale T_{BURST} are annihilated by mode rejection whenever $T_{BURST} \gg T_{RES}$. On the contrary, the system seems to be transparent to these fluctuations when $T_{BURST} \ll T_{RES}$, namely, if the fluctuation of the input signal is too fast, then mode rejection cannot take place effectively. However, these are preliminary results, and we are still unsure about their universal validity.

In conclusion, given the ambiguity of the definition of system response time mentioned above, and given the impossibility to perform any experiment with the current setup, so far we cannot draw firm conclusions about the dynamic of the transient regime or of fast fluctuations in the fibres under test. Therefore, we prefer not to provide data on this regard, as they may be contradicted by more in-depth measurements in the future.

REVIEWERS' COMMENTS

Reviewer #1 (Remarks to the Author):

I thank the authors for the clarification dealing with the response time of the system. I fully understand that additional data would be difficult to include in this article, since they would require a novel and complete investigation. However, I found this discussion on temporal response quite important for applications. So I would recommend to the authors to add at least a short comment in the main manuscript or in the supplements regarding the response time of the system "Tres", which compared to single mode fibres [Kozlov, et al., JOSA B 28, 1782 (2011)], has to be evaluated using an averaged Kerr coefficient, in order to take into account for the multimode nature of the setup.

REVIEWER COMMENTS

Reviewer #1 (Remarks to the Author):

I thank the authors for the clarification dealing with the response time of the system. I fully understand that additional data would be difficult to include in this article, since they would require a novel and complete investigation. However, I found this discussion on temporal response quite important for applications. So I would recommend to the authors to add at least a short comment in the main manuscript or in the supplements regarding the response time of the system “Tres”, which compared to single mode fibres [Kozlov, et al., JOSA B 28, 1782 (2011)], has to be evaluated using an averaged Kerr coefficient, in order to take into account for the multimode nature of the setup

We agree with the reviewer and as suggested we have added a comment in the Supplementary section 2, namely:

“... It is worth noting that these outcomes hold true even in the case where the input FS fluctuates in time, provided that the time scale of the fluctuations is longer than the characteristic response time of the system [1], which here can be roughly approximated with $c/(\gamma_{\text{avg}} \cdot P_f)$, γ_{avg} being the average Kerr coefficient....”

Where [1] is Kozlov, et al., JOSA B 28, 1782 (2011),